# Glut-3 Gene Knockdown as a Potential Strategy to Overcome Glioblastoma Radioresistance

**DOI:** 10.3390/ijms25042079

**Published:** 2024-02-08

**Authors:** Gaia Pucci, Luigi Minafra, Valentina Bravatà, Marco Calvaruso, Giuseppina Turturici, Francesco P. Cammarata, Gaetano Savoca, Boris Abbate, Giorgio Russo, Vincenzo Cavalieri, Giusi I. Forte

**Affiliations:** 1Institute of Molecular Bioimaging and Physiology (IBFM)-National Research Council (CNR), Cefalù Secondary Site, C/da Pietrapollastra-Pisciotto, 90015 Cefalù, Italy; gaia.pucci@ibfm.cnr.it (G.P.); valentina.bravata@ibfm.cnr.it (V.B.); marco.calvaruso@ibfm.cnr.it (M.C.); francesco.cammarata@ibfm.cnr.it (F.P.C.); giorgio.russo@ibfm.cnr.it (G.R.); giusi.forte@ibfm.cnr.it (G.I.F.); 2Department of Biological, Chemical and Pharmaceutical Sciences and Technologies (STeBiCeF), University of Palermo, Viale delle Scienze Bld.17, 90128 Palermo, Italy; g.turturici05@libero.it; 3Radiation Oncology, ARNAS-Civico Hospital, 90100 Palermo, Italy; gaetano.savoca@arnascivico.it (G.S.); borisfederico.abbate@arnascivico.it (B.A.)

**Keywords:** glioblastoma, radioresistance, chemical hypoxia, gene knockdown

## Abstract

The hypoxic pattern of glioblastoma (GBM) is known to be a primary cause of radioresistance. Our study explored the possibility of using gene knockdown of key factors involved in the molecular response to hypoxia, to overcome GBM radioresistance. We used the U87 cell line subjected to chemical hypoxia generated by CoCl2 and exposed to 2 Gy of X-rays, as single or combined treatments, and evaluated gene expression changes of biomarkers involved in the Warburg effect, cell cycle control, and survival to identify the best molecular targets to be knocked-down, among those directly activated by the HIF-1α transcription factor. By this approach, *glut-3* and *pdk-1* genes were chosen, and the effects of their morpholino-induced gene silencing were evaluated by exploring the proliferative rates and the molecular modifications of the above-mentioned biomarkers. We found that, after combined treatments, *glut-3* gene knockdown induced a greater decrease in cell proliferation, compared to *pdk-1* gene knockdown and strong upregulation of *glut-1* and *ldha*, as a sign of cell response to restore the anaerobic glycolysis pathway. Overall, *glut-3* gene knockdown offered a better chance of controlling the anaerobic use of pyruvate and a better proliferation rate reduction, suggesting it is a suitable silencing target to overcome radioresistance.

## 1. Introduction

Glioblastoma (GBM) is the most aggressive form, and it represents approximately 45% of all central nervous system (CNS) cancers. Based on the CNS tumor classification, GBM is classified by the World Health Organization (WHO) as a Grade IV astrocytoma, due to its high invasiveness and capability to infiltrate rapidly into the surrounding brain parenchyma [1]. Concerning other cancer histotypes, GBM has a low incidence, and it accounts for 3–4 cases per 100,000 individuals each year. Both the prognosis and treatment of GBM still represent a real challenge. In fact, despite the increased knowledge of GBM molecular and biological features, the clinical course for patients remains dismal, with a median survival time of 12–15 months and a 5-year survival rate of less than 5% [2]. The current therapeutic strategy for GBM treatment is based on a trimodal approach, which includes tumor surgical resection followed by radiotherapy (RT) combined with adjuvant temozolomide (TMZ) chemotherapy. However, tumor resistance to these therapies and tumor recurrence often occur; hence, the need to develop new and more efficient therapeutic approaches is of primary importance [3]. 

The aggressive nature of GBM depends on its complex pathophysiology, which involves different cellular and molecular mechanisms. On the other side, tumor microenvironment (TME) also plays a key role in GBM progression. TME is characterized by a miscellaneous population of tumor cells, immune cells, and stromal cells interacting in a dynamic and intricate manner. This cellular heterogeneity, alongside tumor-driven angiogenesis, hypoxia, and aberrant signal transduction, contributes to immune evasion, therapeutic resistance, and tumor recurrence [4,5]. 

From a molecular point of view, GBM is featured by genomic instability, resulting in several genetic and epigenetic aberrations such as amplification and mutation of the *epidermal growth factor receptor* (*egfr*), loss of function of *the phosphatase and tensin homolog* (*pten*), mutations in the *isocitrate dehydrogenase 1* (*idh1*), and *tumor protein p53* (*tp53*) genes. These alterations trigger the activation of the molecular pathways of PI3K/AKT/mTOR and RAS/RAF/MEK/ERK, thus leading to uncontrolled cell proliferation, evasion of apoptosis, and tissue invasion [3,5,6].

Recently, DNA methylation, histone modifications, and the expression of non-coding RNA have been highlighted as epigenetic dysregulations occurring in GBM [7]. One example is the methylation of the promoter of the *O6-methylguanine-DNA methyltransferase* (*mgmt*) gene, which may alter the response to TMZ treatment [8]. Moreover, aberrations in both microRNAs and long non-coding RNAs transcription have been found to regulate the proliferation, migration, and invasion of GBM cells, as well as the modification of the TME [8,9].

In addition, hypoxic conditions promote the malignant progression of GBM [10]. A typical behavior of cancer cells is an altered metabolism, i.e., an increase in glucose uptake and anaerobic respiration, which causes the fermentation of glucose to lactate [11]. This phenomenon, known as the Warburg effect, occurs also in the presence of fully functioning mitochondria [12,13]. Hypoxia is very common in several cancers and represents a trait of a bad prognosis. Indeed, it promotes angiogenesis and vasculogenesis, modifies the tumor cell metabolism, causes genomic instability, and promotes the development of cancer stem cells (CSCs) and circulating tumor cells that play a key role in metastasis formation [14]. All these elements contribute to generating chemoresistance and radioresistance [15]. Indeed, the tumor is protected by the hypoxic microenvironment, which is, therefore, an adverse risk factor for the RT clinical outcome, since hypoxic tumors need a higher dose of radiation to obtain an efficient cell-killing rate than normoxic ones. 

The hypoxic pattern of GBM has been well documented and is one of the primary causes of radioresistance [16]. According to the oxygen fixation theory, under normoxia, molecular oxygen favors permanent DNA damage, caused by reactive oxygen species (ROS), generated during water radiolysis (indirect impacts of ionizing radiation) [15]. On the other hand, when the concentration of oxygen is dramatically reduced, as in hypoxia, the detrimental effects induced by ROS are proportionately decreased, affecting indirect RT damage and establishing the so-called GBM radioresistance [17].

Therefore, hypoxia-induced radioresistance is one of the major challenges in the treatment of GBM, whereas one of the main goals of radiobiology research is thus to highlight the mechanisms inducing radioresistance in GBM and to develop new approaches to overcome it. 

We recently used the U87 cell line, as a model of GBM, to study the radiosensitizing effects of a SRC family kinase inhibitor [18,19]. We also determined the influence of physical hypoxia on the activation/inhibition of specific pathways that confer radioresistance to proton therapy under low oxygen concentrations [15].

In this work, we created an in vitro system that could have radiosensitizing effects on U87 cells under chemical hypoxic conditions. For this purpose, we used cobalt chloride (CoCl_2_), a chemical hypoxia-mimicking agent widely employed to induce a “hypoxia-like” condition by stabilizing *hypoxia-inducible factor 1* (HIF-1α), the master regulator of the cell adaptive response to hypoxia [20]. The use of CoCl_2_ to create chemical hypoxia has a significant advantage since it is a smoother system that is not affected by the technical complexity required to generate physical hypoxia. The HIF-1α transcriptional factor is activated when the oxygen concentration at the cellular level decreases. Hence, HIF-1α acts as a sensor of oxygen concentration and it represents the master regulator of gene expression during hypoxia, by directly and indirectly regulating numerous genes, involved in DNA repair and in conferring metabolic adaptation to hypoxic conditions [11]. In particular, through the binding to specific *hypoxia-response element* (HRE) consensus (5′-ACGTG-3′) on promoters, the HIF transcription factor directly regulates different genes, adapting cells to survive under hypoxia conditions [11]. It has been reported that HIF-1α actively participates in metabolism regulation, inducing the switch towards anaerobic glycolysis, for example, upregulating the glucose transporters (GLUTs) and *pyruvate dehydrogenase kinase 1* (PDK-1) [21]. While GLUT carriers are responsible for the glucose uptake inside cells, which is the first glycolysis step, PDK-1 prevents pyruvate from entering the Krebs cycle and mediates the conversion of pyruvate to lactate [22]. As a result, HIF-1α leads to the shift from oxidative phosphorylation to anaerobic glycolysis, thus stimulating the Warburg effect [23]. Our work was aimed at identifying key genes upregulated by HIF in U87 cells subjected to radiations under hypoxia conditions, to test their silencing as a radiosensitization method to overcome radioresistance. 

## 2. Results

### 2.1. Effects of CoCl_2_ on U87 Cell Proliferation and Morphology

In order to mimic hypoxia in the U87 cell line, we used CoCl_2_, one of the most widely used agents able to induce chemical hypoxia because it stabilizes HIF-1α/2α under normoxic conditions [20]. U87 cells were subjected to CoCl_2_ treatment at concentrations of 50 µM and 100 µM, and the evaluation of the cell proliferation was carried out at 24 h, 48 h, and 72 h after treatment by means of cell counting with Burker chamber. As shown in Figure 1A, the 24 h post-treatment cell proliferation trend with both CoCl_2_ concentrations was similar, with a decrease of 28% and 35% in 50 µM- and 100 µM-treated samples, respectively, compared to the controls. On the contrary, cell toxicity increased at later times with a significant decrease in cell proliferation of 69% and 77% with 50 µM and 100 µM at 48 h and of 82% (*p*: 0.01) and 89% (*p*: 0.007) with 50 µM and 100 µM at 72 h, respectively.

We also evaluated CoCl_2_’s effects on cell morphology by phase-contrast microscopy observations during treatment times. Twenty-four hours after treatment, the cell morphology was regular and comparable to untreated cells (Figure 1B), while morphological variations, characterized by a flattened shape and cytoplasmic extensions, signs of cell stress, were visible at 48 and 72 h after treatments with both concentrations of CoCl_2_. 

Thus, hereafter 18 h of CoCl_2_ treatment has been used to further reduce cytotoxicity. 

### 2.2. Analysis of Molecular Response to Induced Hypoxia through Specific HIF-1ɑ Downstream Biomarkers 

The Hif-1α transcription factor is responsible for the activation of genes involved in the intensification of anaerobic respiration (Warburg effect), cell cycle control, and cell fate balance inhibition, thanks to the presence of HRE recognition sequences on their promoters in single or multiple copies. Thus, to identify a suitable candidate biomarker for gene silencing, we selected the following genes directly activated by HIF-1α: *glut-1*, *glut-3*, *ldha*, *eno1*, *pdk-1*, and *lon* as metabolic markers of the Warburg effect [11,22,23]; *lncRNA-p21* and *miR-210* as hypoxia regulation markers [24,25]; p21 as cell cycle control marker [26]; *Survivin*, *Livin*, and *mir-590* as survival markers [27,28,29]. These genes were tested for gene expression changes under hypoxia using 50 µM CoCl_2_ or radiation treatment using photon beams with the dose of 2 Gy, as single treatments or in combination (Table 1). In particular, a slight increase in *hif-1α* gene expression was observed under the three described treatments (1.05-, 1.22-, and 1.38-fold changes, respectively), confirming its involvement in inducing radioresistance in hypoxic conditions. A high increase in *glut-1* and *glut-3* expression was obtained under hypoxia, with 2.10- and 4.05-fold changes, respectively, whereas a moderate increase of 1.92- and 1.62-fold resulted after 2 Gy/hypoxia combined treatment, respectively. Similarly, an increased expression of *eno1*, *ldha*, and *pdk-1* genes was observed under hypoxia as single or combined treatment with 2 Gy radiation (*eno1*: 1.78 and 1.37; *ldha*: 1.43 and 1.41; *pdk-1*: 2.06- and 2.59-fold changes, respectively). Particularly, the *ldha* and *pdk-1* increased expression suggested a metabolic switch towards anaerobic respiration, as both these genes contribute to favor the pyruvate subtraction from the Krebs cycle towards lactate conversion [11]. Overall, the above-mentioned genes were not overexpressed following the treatment with only radiation. Furthermore, a slightly increased expression was also obtained for the gene encoding the LON protease (1.30- and 1.16-fold changes under hypoxia alone or in combination, respectively), which degrades COX4-1, the cytochrome oxidase isoform predominant in normoxia, so that HIF-1 may fine-tune mitochondrial respiration by upregulating the COX4-2, more efficiently in hypoxia. In addition, an upregulation was also observed for *lincRNA-p21* and *miR-210* under hypoxia (1.50- and 1.36-fold change), but not under 2 Gy radiation or combined treatment. Their activation represents a specific hypoxia signature, as LincRNA-p21 promotes the *hif-1α* upregulation through a positive feedback effect and prevents its degradation, whereas miR-210 is considered a multi-target factor of hypoxia response [24,25]. 

The *p21* gene, instead, was upregulated under all the three treatments tested (2.16-, 1.29-, 2.38-fold changes under hypoxia, 2 Gy radiation, and their combination, respectively), as it represents a crucial factor controlling checkpoint for cell cycle arrest and cell fate [26]. 

On the other hand, *livin* and *survivin* were both downregulated under hypoxia and radiation treatment alone or combined, instead a consistent upregulation was observed for *miR-590* (3.51- and 2.03-fold changes under hypoxia and combined treatment, respectively), which is known to inhibit apoptosis by downregulating Hif-1α [29]. 

Based on the above-described gene expression changes in response to hypoxia induced by 50 µM CoCl_2_, *glut-3* and *pdk-1* genes were chosen as targets for gene knockdown.

### 2.3. Radiosensitizing Effects of glut-3 and pdk-1 Morpholino-Induced Gene Knockdown

To study the radiosensitizing effects related to morpholino-induced gene silencing, the U87 cells were subjected to different treatment conditions, as reported in the Section 4, and cell proliferation was evaluated using cell counting with Burker chamber at 7, 11, and 15 days after treatments (Figure 2). 

First, we tested the toxicity caused by the silencing system per se, i.e., using Mo-ST at 7.5 μM. As shown in Figure 2A, Mo-ST treatment led to cell proliferation values almost comparable to those of the untreated controls at all three time points analyzed, with a reduction value of 11% and 7% at 7 and 11 days post-treatment, respectively. Any variation was observed at 15 days.

Secondly, we evaluated the effects induced by 7.5 μM Mo-ST, 7.5 μM Mo-Glut3, and 7.5 μM Mo-PDK-1 under hypoxic conditions without radiation. As shown in Figure 2B, the cell proliferation trend was similar among the three condition treatments, with reduction values between 16% and 18% at 7, 11, and 15 days post Mo-Glut3 treatment and between 5% and 17% at the same time points post Mo-PDK-1 treatment.

Finally, following combined treatments with 2 Gy X-rays under hypoxic conditions, Mo-Glut3 induced a greater decrease in cell proliferation compared to Mo-PDK-1, with reduction values between 69% (*p*: 0.04) and 64% (*p*: 0.001), at 7, 11, and 15 days, and between 57% (*p*: 0.001) and 55% (*p*: 0.009), at the same time points, respectively (Figure 2C).

### 2.4. Molecular Response in U87 Cells under CoCl_2_-Induced Hypoxia Condition and glut-1/pdk-1 Gene Knockdown 

To assess the molecular effects of *glut-3* or *pdk-1* gene silencing as a radiosensitizing method to overcome radioresistance induced by hypoxia, we assayed the expression of four key genes involved in the induction of Warburg effect (*glut-1*, *glut-3*, *ldha*, and *pdk-1*) and four key genes involved in the survival/death balance (*survivin*, *bax*, *bcl-2*, and *casp-9*) (Figure 3). They were assayed in U87 cells, following treatments with 50 µM CoCl_2_, 2 Gy of IR, and Morpholinos (Mo-GLUT-3 or Mo-PDK-1) alone and in double or triple combinations. Table 2 presents up- and down-regulated genes. 

Regarding the Warburg effect evaluation, as already reported in Table 1, under hypoxia condition (50 µM CoCl_2_) alone or in combination with 2 Gy of radiation, an increased gene expression of *glut-1*, *glut-3*, *ldha*, and *pdk-1* was observed, a sign of a metabolic switch toward anaerobic glycolysis. 

However, the gene silencing using Mo-PDK-1 induced a stronger upregulation of *glut-1* and *ldha* under hypoxia conditions both in the double (4.92- and 2.31-fold changes, respectively) and triple combined treatment (8.82- and 2.34-fold changes, respectively), as a sign of cells response to restore the metabolic pathway.

On the other hand, the use of Mo-GLUT-3 induced upregulation of *glut-1*, *pdk-1*, and *ldha* in double combination with hypoxia (2.91-, 1.84-, and 1.50-fold changes, respectively). Instead, in the triple combined treatment, including 2 Gy of IR, although it favored a *glut-1* increased expression under hypoxia condition, as compensation for glucose entry into the cells, it induced only a moderately increased expression of *ldha* but not of the *pdk-1* gene (1.32- and 0.92-fold changes, respectively). 

Thus, regarding this metabolic aspect, the *glut-3* gene silencing seems to offer a better chance of moderating the use of anaerobic glycolysis in U87 hypoxic cells subjected to 2 Gy of radiation.

Instead, considering the death/survival balance, the hypoxia condition and 2 Gy radiation per se did not produce very high variations in *bax*, *casp-9*, *bcl-2*, and *survivin* expression, especially if we considered the combined treatment (0.95-, 0.68-, 1.20-, 0.97-fold changes, respectively). However, the *pdk-1* gene silencing caused increased expressions of both pro-survival and pro-apoptotic signals, with greater expression levels of pro-survival *bcl-2* and *survivin* genes in the triple combined treated cells (2.30- and 1.75-fold changes, respectively), with respect to *bax* and *casp-9* (1.38- and 1.05-fold changes, respectively). Similarly, in *glut-3* gene-silenced U87 cells, both hypoxia and 2 Gy of IR were able to upregulate *casp-9* but not *bax* as pro-apoptotic signals and *bcl-2* and *survivin* as pro-survival genes. However, in the triple combined treatment, this balance leaned towards survival signals with greater expression levels of pro-survival *bcl-2* and *survivin* genes (3.15- and 2.34-fold changes, respectively), with respect to *bax* and *casp-9* (1.60- and 1.63-fold changes, respectively). 

Thus, regarding the cell fate balance, both the *pdk-1* and *glut-3* genes silencing induced an early survival tentative, evaluated 18 h post hypoxia treatment. 

## 3. Discussion

GBM care is one of the major clinical challenges in oncology. It is featured by a certain rate of radioresistance, which is caused by well-known mechanisms, such as the presence of hypoxic regions and consequent metabolic adaptation to oxygen reduction [30]. Indeed, irregular blood vessels characterize the GBM microenvironment, creating niches where cycling hypoxia occurs and several molecular signals are activated by the HIF pathway, such as stem cell renewal and reprogramming [31,32,33,34]. 

Numerous strategies have been tested until now, starting from the research in clinical RT, with aims of improving treatment plans, both in terms of technological advancement of beam delivery on the brain [35,36,37] or with clinical trials aimed at testing new dose hypofractionation protocols [38,39] or reirradiation of recurrent GBM [40]. However, despite efforts in RT clinical research, the radiation treatments remain insufficient to overcome resistance, especially of recurrent GBMs. 

Many other approaches have been explored. One was concerned with the reduction of hypoxia in GBM through intratumoral oxygenation to increase glioma radiosensitivity [41,42], while another focused on using imaging methods to identify hypoxic regions where to deliver escalated radiation doses [43]. 

In addition, several studies have been published, about treatments that use targeted molecules or radiosensitizers [30]. The main pathways chosen to develop targeted or inhibitor molecules are those already known to be implicated in the GBM progression and radio/chemoresistance mechanisms, such as DNA damage repair (DDR) machinery, Notch, PI3-kinase/Akt/mTOR, JAK/STAT, and Wnt signalings [15]. Some of them have been tested as radiosensitizers in the GBM treatment, to amplify the effectiveness of RT without dose increasing and sparing healthy tissues. In particular, oxygen carriers, such as hemoglobin or fluorocarbons [42], or oxygen-mimicking compounds have been investigated for their role in trying to overcome local hypoxia in GBM tumors. Among these, the trans sodium crocetinate (TSC) enhances the oxygen delivery to hypoxic tissues, showing no toxic effect in Phase I and II clinical trials (NCT01465347), but without significantly increasing survival [44]. Other approaches, most frequently explored, regarded the use of pro-oxidant molecules, chemotherapeutic drugs, or DDR-interfering molecules. In addition, numerous trials have been conducted to find effective radiosensitizers, most failed to show an enhancement of progression-free or overall survival, and some others are ongoing with encouraging results [30]. 

Our study aimed to explore the possibility of using gene silencing of key factors involved in the molecular response to hypoxia, to overcome GBM radioresistance. In a previous study, we compared survival curves of irradiated normoxic vs. hypoxic (0.2% O_2_) U87 cells with increasing doses of a proton beam, finding a substantial increase in cell survival in hypoxic conditions with an oxygen enhancement ratio (OER)_S=10%_ = 1.69 ± 0.36 [15]. Furthermore, we analyzed whole genome gene expression changes observed in 2 Gy and 10 Gy hypoxic U87 cells with respect to untreated U87 cells or with respect to irradiated normoxic U87 cells with the same doses. Thus, we reconstructed top-statistically deregulated pathways, exclusively activated by hypoxia (dose-independent) (i.e., comparison 2 Gy hypoxia vs. 2 Gy normoxia) or activated by combined treatment (radiation/hypoxia), compared to untreated U87 cells. Overall, dose-independent signals that were activated by hypoxia included platelet activation, Wnt and Ras signaling, proteoglycans in cancer, and protein processing in the endoplasmic reticulum, while other pathways were affected by the administered dose of 2 or 10 Gy or by their combination with the oxygen level [15]. 

However, in this study, we would select specific candidate biomarkers to be silenced, choosing them among a list of genes directly regulated by HIF transcription factor, for the presence of HRE consensus sequence on their sequence promoters. In particular, they were *glut-1*, *glut-3*, *ldha*, *eno1*, *pdk-1*, *lon* as metabolic biomarkers of Warburg effect [11,22,23]; *lincRNA-p21* and *miR-210* as hypoxia regulation biomarkers [24,25]; p21 as cell cycle control biomarker [26]; *survivin*, *livin*, and *miR-590* as survival biomarkers [23,27,29].

This time we used CoCl_2_ to create chemical hypoxia, which has the advantage of being a simple system to uniformly treat cells in their normal supports, without the necessity of using smaller chambers and oxygen sensors. In addition, the treatment with CoCl_2_ has been tested for toxicity in terms of concentrations and kinetics of treatment, revealing a low toxicity level within 24 h using 50 µM CoCl_2_, with a non-significant cell proliferation reduction (28%) and normal cell morphology, compared to U87 untreated cells. Thus, we decided to maintain the treatment window up to 18 h, largely within the 24 h window tested, as the objective was to create cell adaptation to hypoxia, without compromising their long-term vitality. 

Furthermore, we assessed the correct hypoxia establishment on two fronts, directly, by highlighting fluorescent nuclei due to the expression of the recombinant HIF protein from the HIF1α-GFP transgene transfection and, indirectly, by evaluating gene expression activation of its downstream, above-mentioned, HRE-regulated genes. Both these approaches demonstrated the hypoxia instauration in our in vitro model. 

Thus, in order to choose candidate genes to be silenced, we compared the gene expression variations of the HRE-regulated genes list under 50 µM CoCl_2_ or 2 Gy of photon beam, as a single or combined treatment (Table 1).

Overall, genes involved in the metabolic response to oxygen reduction were upregulated in hypoxic samples, treated as a unique treatment or in combination with radiation, and they were not only in irradiated samples. High levels of *glut-1* and *glut-3* expression were revealed under hypoxia and a moderate increase resulted after the 2 Gy/hypoxia combined treatment, respectively, suggesting the glycolysis upregulation. Similarly, increased expression of *eno1*, *ldha*, and *pdk-1* genes was observed under hypoxia as a single or combined treatment with 2 Gy radiation. In particular, the *ldha* and *pdk-1* increased expression were clear signs of a metabolic switch towards anaerobic respiration, as both these genes contribute to favoring the pyruvate subtraction from the Krebs cycle towards lactate conversion. Moreover, the anaerobic adaptation hypothesis was reinforced by the observed upregulation of the *lon* protease gene, which is known to degrade the cytochrome oxidase isoform predominant in normoxia (COX4-1), in favor of the HIF-mediated transcription of a more efficient hypoxic form (COX4-2). 

In addition, other two typical markers of the HIF signature, the *lincRNA-p21* and *miR-210*, were found to be upregulated under hypoxia, but not in samples treated with 2Gy radiation or combined treatment. Nonetheless, *lincRNA-p21* promotes the *hif-1α* upregulation through a positive feedback effect, also preventing its degradation, and *miR-210* is a multi-target factor of hypoxia response; these two genes were unsuitable candidates for gene silencing, as they were not upregulated in the combined treatment with radiations [24,25].

Regarding the cell fate balance, we found upregulation of the *p21* gene as a crucial checkpoint factor controlling cell cycle arrest and repair [26]. It was upregulated in all the three treatments tested. On the other hand, *livin* and *survivin*, as survival signals directly regulated by HIF, were both downregulated by hypoxia and radiation treatment alone or combined; instead, a consistent upregulation was observed for *miR-590* [27,28,29]. The last one is known to inhibit apoptosis by downregulating Hif-1α.

However, considering the good concordance of gene expression among metabolic genes in response to hypoxia and its combination with radiation, we have chosen to perform gene silencing of 2 genes, one at the beginning and one at the final part of the glycolysis. In particular, we have chosen to silence *glut-3*, both for the large upregulation observed and because it is a specific neuron isoform. Indeed, in a prospective clinical application, a radiosensitization method aiming at silencing *glut-3* could be translated into a specific treatment on brain areas that greatly increase its expression, i.e., the hypoxic tumor zones [45]. Furthermore, we also evaluated the *pdk-1* silencing, as a key gene in the metabolic switch towards the anaerobic utilization of pyruvate. In this way, blocking anaerobic glycolysis from two different fronts, we tried to understand how the metabolic response changes following irradiation in hypoxic conditions. 

For this purpose, we used Vivo-morpholino oligonucleotides to obtain knockdown of the *glut-3* and *pdk-1* genes, as better described in the methods section. This system is featured by a specific antisense-mediated exon skipping of pre-mRNA, covalently linked to an octa-guanidine dendrimer as a delivery chain, which has been shown to efficiently penetrate cells after systemic or local injection in preclinical models, thus representing a ready-to-use system in case of preclinical future experimentation.

Then, after setting no toxic range of Mo-concentration using Mo-ST as the negative control, we investigated the radiosensitizing power of GLUT-3 and PDK-1-Mo. Figure 2 shows cell proliferation variations in three configurations of comparison. Negligible proliferation rate reductions were imputable to the morpholino per se treatment after 7, 11, and 15 days post-treatment with 7.5 μM Mo-ST (Figure 2A). On the other hand, as shown in Figure 2B, a reduced proliferative trend has been observed after treatments with 7.5 μM Mo-Glut3 and 7.5 μM Mo-PDK-1, with respect to 7.5 μM Mo-ST, under hypoxic conditions, with reduction ranges between 16% and 18% in the 7–15 time window days post Mo-Glut3 treatment and between 5% and 17% at the same time window, post Mo-PDK-1 treatment. However, as shown in Figure 2C, considering as reference the U87 cell count after 7, 11, and 15 days post-combined treatment with 2 Gy X-rays in hypoxic conditions, a significant reduction in the proliferative rates was observed in the triple combination with Mo-Glut-3 and Mo-PDK-1. In particular, the radiosensitizing effect produced by the Glut-3 silencing was higher, as it induced a greater decrease in cell proliferation, with values between 69% (*p*: 0.04) and 64% (*p*: 0.001), with respect to reduction values of 57% (*p*: 0.001) and 55% (*p*: 0.009) induced by Mo-PDK-1, at 7–15 days of the time window, respectively (Figure 2C).

Then, we proceeded to analyze the molecular effects of *glut-3* or *pdk-1* gene silencing as a radiosensitizing method. To this aim, we assayed gene expression variation in four key genes involved in the Warburg effect induction (*glut-1*, *glut-3*, *ldha*, and *pdk-1*) and four key genes involved in the survival/death balance (*survivin*, *bax*, *bcl-2*, and *casp-9*) in the U87 cells treated with 50 µM CoCl_2_, 2 Gy X-rays, and Morpholinos (Mo-GLUT-3 or Mo-PDK-1), alone and in double or triple combinations. Table 2 presents up- and down-regulated genes. 

In particular, the gene silencing using Mo-PDK-1 induced a strong upregulation of *glut-1* and *ldha* under hypoxia conditions both in the double and triple combined treatments, as a sign of cell response to restore the anaerobic glycolysis pathway.

On the other hand, the use of Mo-GLUT-3 induced upregulation of *glut-1*, *pdk-1*, and *ldha* in double combination with hypoxia. However, in the triple combined treatment, including 2 Gy X-rays, although it favored a *glut-1* increased expression under hypoxia condition, as compensation for glucose entry into the cells, it induced only a moderately increased expression of *ldha* but not of the *pdk-1* gene. Figure 3 summarizes the molecular network involving the above-reported genes activated by hypoxia. 

Thus, the *glut-3* gene silencing seems to offer a better chance of interfering with the anaerobic use of pyruvate in U87 hypoxic cells subjected to 2 Gy of radiation.

In addition, the combined treatment of hypoxia and 2 Gy X-rays did not produce considerable variations in the *bax*, *casp-9*, *bcl-2*, and *survivin* expression (Table 2). However, the cell fate balance was affected in the triple combinations, toward a pro-survival cell reaction. Particularly, the *pdk-1* gene silencing caused an increased expression of both pro-survival and pro-apoptotic signals, with greater expression levels of pro-survival *bcl-2* and *survivin* genes in the triple combined treated U87 cells, with respect to *bax* and *casp-9*. 

Similarly, in *glut-3* gene-silenced U87 cells, subjected to the triple combined treatment, this balance shifted towards survival signals with greater expression levels of pro-survival *bcl-2* and *survivin* genes, with respect to *bax* and *casp-9*. These results regarding the survival/death balance were also in line with gene expression profiling experiments previously performed by our group on the U87 cells subjected to physical hypoxia and 2 Gy of IR [15]. 

Thus, both the PDK-1 and GLUT-3 genes silencing induced an early survival tentative, evaluated a few hours post hypoxia treatment, even if the proliferative trend studied at 7, 11, and 15 days after triple treatments, showed in both cases significant proliferative rate reduction, as a sign of the radiosensitization ability of these two antisense mRNAs.

Overall, this study showed the *glut-3* and *pdk-1* gene silencing as a suitable and effective radiosensitizing method to overcome resistance induced by the Warburg effect instauration in hypoxia condition, typical of some areas of GBM tumor. 

The metabolic switch toward high rates of glycolysis has been correlated with many instances of cancer resistance, and its inhibition leads to sensitization; thus, many authors dedicated their efforts to sensitize cancer by inhibiting the Warburg effect [46,47,48,49,50,51,52,53,54,55]. 

Fewer authors applied this strategy to GBM [56,57], whereas some others have explored the possibility of combined treatments with RT in other forms of cancer, but not in GBM [54,58,59]. In this regard, encouraging results derived in 2016 from the study of Vartanian A. et al., who have targeted *hexokinase 2* (*hk2*) to sensitize U87 and primary GBM cells and a xenograft GBM model subjected to combined radio-chemotherapy with TMZ treatment [60]. Overall, the triple combination including *hk2* knockdown resulted in a significant improvement in overall survival concerning the experimental arm of radio-chemio double treatment. In particular, the Warburg effect inhibition was assessed for the HK2 loss, but not for HK1 or HK3. Furthermore, the radiosensitization was mediated by increased DNA damage, via activation of ERK signaling. However, HK2 is also expressed by normal muscle cells; thus, systemic toxicity could be expected by developing systemic HK2 inhibitors.

Thus, our study would cover a literature gap, showing cell and molecular modification induced by the silencing of key genes controlling the metabolic anaerobic switch in triple combined treated U87 GBM cells, as a radiosensitization method. Moreover, in our study, *glut-3* silencing offered a better chance of controlling the anaerobic use of pyruvate and a better proliferation rate reduction. This is a promising result, as GLUT3 is considered the main, even not the exclusive neuronal glucose transporter; whereas in other tissues, it is less prevalently expressed, also in sperm, embryos, and white blood cells, together with the other transporters, which could supply *glut-3* silencing. In addition, a promising GLUT-3 feature is represented by its upregulation in case of increased energy necessity, as it has a fivefold greater transport capacity than GLUT1 or GLUT4 and higher glucose affinity than GLUT1, GLUT2, or GLUT4 [45]. Thus, the GLUT-3 higher expression in the hypoxic GBM area could represent a way to direct the radiosensitization activity inside the tumor. Future preclinical experimentation will be useful to better highlight its in vivo radiosensitization ability and gain in progression-free or overall survival.

## 4. Material and Methods

### 4.1. Cell Culture, CoCl_2_ Treatments, and Cell Proliferation

The U87 MG human glioblastoma cell line was purchased from the European Collection of Authenticated Cell Cultures (ECACC, Public Health England, Porton Down Salisbury, UK), cultured, and maintained as previously reported [18]. 

To induce chemical hypoxia, the U87 cells were seeded at a density of 35 × 10^4^/flask in T-25 flasks overnight, treated with 50 μM and 100 μM of CoCl_2_, and maintained in culture in standard conditions. Cell proliferation was evaluated by cell counting by using the Burker camera at 24, 48, and 72 h after treatment compared to control samples (untreated cells). The values of cell survival are expressed as the mean ± SD obtained from three independent experiments.

### 4.2. Cytotoxicity Assays

Toxicity effects induced by CoCl_2_, Lipofectamine, and Morpholino oligonucleotides were determined by using the luminescent CytoTox-Glo™ Assay (Promega, Madison, WI, USA), according to the manufacturer’s instructions, seeding the U87 cells (10^4^ cells/well) in 96-well plates the day before treatments. Luminescent signals were detected by the microplate reader Victor3 (PerkinElmer, Waltham, MA, USA). Cell viability was calculated by subtracting the luminescent dead-cell signal from the total luminescent value and normalizing data to the total cell number. 

### 4.3. Irradiation and Combined Treatments

Cell irradiation was performed with photon beams (X-rays) of 6 MV nominal energy using the dose of 2 Gy, using a medical linear accelerator (Siemens Medical Systems, Concord, CA, USA). The Linac calibration, irradiation setup, and dose distribution were conducted as previously described [61]. For combined treatments, the experimental workflow was as follows: day 1: U87 cells seeding; day 2: cell treatment with 7.5 μM Mo-ST or 7.5 μM Mo-Glut3 or 7.5 μM MoPDK-1; day 3: medium change and treatment with 50 μM CoCl_2_ for 18 h; day 4: cell irradiation with 2 Gy X-rays; day 5: total RNA extraction and qRT-PCR (Figure 4). To evaluate cell proliferation, after 18 h of combined treatments, the medium was changed, cells were left in culture to grow under standard conditions, and cell counting was performed at 1 week, 11, and 15 days.

### 4.4. Optical and Fluorescence Microscopy

Cell observation and evaluation were carried out both with optical microscopy by using a phase-contrast microscope (CarlZeiss, Göttingen, Germany) and by a fluorescence inverted microscope (Leica DMi8, Wetzlar, Germany). Images and nuclear fluorescence mean density data were acquired by using Leica Application Suite X software, version 5.1.0 (LAS X).

### 4.5. HIF1α-GFP Transgene Construction and Transfection 

To assess the correct hypoxia establishment in our cellular model, we transfected the U87 cells with the HIF-1α-GFP transgene, in which the HIF-1α cDNA was joined in frame with the GFP coding sequence. In detail, the Hif-1α cDNA coding sequence has been amplified, by using sequence-specific primers reported in Table 3, bearing the cleavage sites for the restriction enzymes NheI and BamHI and the following amplification program (94 °C 30″ followed by 35 cycles of 94 °C 30″, 55 °C 30″, 68 °C 3′). Then, the amplicon was purified using the PureLink PCR Purification kit (Thermo Fischer Scientific, Waltham, MA, USA). After restriction digestion of the amplicon and the pEGFP-N1 plasmid (Addgene, Watertown, MA, USA), with NheI (New England BioLabs, Ipswich, MA, USA) and BamHI (Amersham Biosciences, Piscataway, NJ, USA) restrictions enzymes, according to the manufacturer’s instructions, the Rapid DNA Dephos and Ligation kit (Roche, Basel, Switzerland) have been used to perform dephosphorylation and ligation, considering the 1:4 molar ratio of insert to carrier, for a total amount of 120 ng DNA. The transformation occurred in *E. coli* Top10F cells using 1/10 of the ligase reaction volume in a final volume of 100 μL. Kanamycin-resistant colonies were selected on LB agar medium and picked to perform PCR colony using DyNAzyme™ II DNA Polymerase (Thermo Fisher Scientific, Waltham, MA, USA) and Hif-1α-specific primers, to identify transformed colonies containing the HIF1α-GFP transgene. 

Then, transfection of the plasmid leading to the Hif-1α-GFP transgene was performed in U87 cells, according to the Lipofectamine^®^ 2000 Reagent protocol (Thermo Fisher Scientific, Waltham, MA, USA). In particular, 1.2 × 10^5^ cells were cultured in 12 multiwell plates in 1 mL of medium. After 24 h, the transfection was performed using 1 μL/mL of transfecting agent and 0.8 μg/mL of plasmid DNA, which showed the lowest toxicity (dead cells: 13.75% compared to 5.3% in control), among other combinations tested using cytotoxicity assays described above. Instead, co–treatment with 1 μL/mL of the transfecting agent, 0.8 μg/mL of plasmid DNA, and 50/100 μM CoCl_2_ leads to 17% and 15.8% of dead cells (Appendix A).

Furthermore, to better verify and detect the HIF1α-GFP fusion protein into the nucleus, fluorescent staining of U87 nuclei was performed under hypoxia generated with 50 μM CoCl_2_, by Hoechst 33342 solution at a concentration of 1.5 μg/mL. The chosen combination of 1 μL/mL of transfecting agent and 0.8 μg/mL of plasmid DNA showed a higher number of GFP-positive cells (2.5×) than untreated cells, overall showing higher nuclear GFP fluorescence intensity (AU) (8×) (Figure 5A,B).

### 4.6. RNA Silencing by Using Vivo-Morpholino

Vivo-Morpholino antisense oligonucleotides were specifically designed to perform in vitro silencing of GLUT-3 and PDK-1 mRNA and were covalently linked to a generic delivery group, which is composed of an octa-guanidine dendrimer (GENE TOOLs, LLC, Philomath, OR, USA). Vivo-Morpholinos freely diffuse between the cytosol and nuclear compartments and bind complementary sequences of RNA. They were designed to skip one exon and are reported in Table 4. In addition, a Vivo-Morpholino Standard (MO-ST) oligonucleotide was also purchased and used as a control of intrinsic nonspecificity and cytotoxicity. They were used on U87 cell culture according to the supplier-suggested protocol. Briefly, the desired concentration was added into the culture medium the day after seeding cells, and 24 h post-incubation, the medium was replaced with a fresh one. MO-ST was used to assess the absence of cell toxicity on U87 cells, using CytoTox-Glo assay. Appendix A reports viability percentage values obtained in untreated U87 cells (87.4%) vs. 15 μM MO-ST treated cells (92.1%), showing no toxicity induced by the in vivo morpholino treatment per se.

Then, the knockdown effect was detected on the mRNA target by RT-PCR (primers sequences reported in Table 4). Based on the exon skipping, the major or exclusive presence of the expected amplicon of 249 bp instead of 410 bp indicated the GLUT-3 mRNA silencing, whereas the major or exclusive presence of the expected amplicon of 276 bp instead of 418 bp indicated the PDK-1 mRNA silencing. 

Based on the results of cytotoxicity and PCR assays, the concentration of 7.5 μM has been chosen for both the GLUT-3 and PDK-1 mRNA knockdown.

### 4.7. Total RNA Extraction, cDNA Synthesis, and Quantitative Real-Time PCR (qPCR)

Total RNA was extracted from 6 × 10^5^ U87 cells subjected to hypoxia, irradiation, and transfection with morpholino oligonucleotides, as single or combined treatments, following TRIzol Plus RNA Purification Kit (Thermo Fisher SCIENTIFIC) protocol. 

The reverse transcription reactions were performed using the SuperScript™ II Reverse Transcriptase (Thermo Fisher SCIENTIFIC), starting from 1 μg of total RNA using random primers. 

Finally, qPCR was performed using the Fast SYBR™ Green Master Mix and sequence-specific primers (Table 3) for the following target genes: *hif-1α*, *glut-3*, *glut-1*, *ldha*, *eno-1*, *pdk-1*, *lon*, *p21*, *survivin*, *livin*, *mir-210*, *mir- 590*, *lincp21*.

### 4.8. Statistical Analysis

Statistical data assessment was performed using GraphPad Instat (Version 3.05). One-way analysis of variance (ANOVA) was used to analyze if variation among variables means is significantly greater than expected by chance. In particular, it was applied to evaluate the mean of the cell counts obtained for the treated samples vs. the control ones at each time point under analysis. Overall, statistical significance was defined at *p* ≤ 0.05.

## Figures and Tables

**Figure 1 ijms-25-02079-f001:**
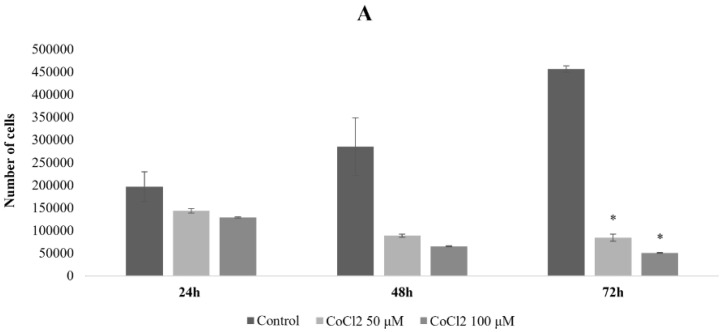
Cell counts of U87 cells untreated and treated with 50 and 100 μM CoCl_2_. On the abscissa axis, the origin for the hours corresponds to the start of the treatment (**A**). Micrographs of U87 cells at 24, 48, and 72 h post-treatment with 50 and 100 μM CoCl_2_ (magnification 10×) (**B**). The data shown are representative of two independent experiments and are expressed as the mean ± standard deviation of the mean (SD). The significance level compared to the control sample, for each time point, was set to *p* < 0.05 (unpaired *t*-test Welch corrected) and displayed with the asterisk (*).

**Figure 2 ijms-25-02079-f002:**
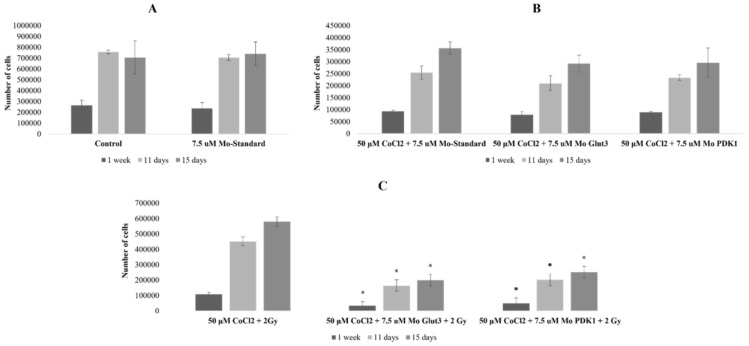
Cell counts of U87 cells untreated vs. treated with 7.5 μM Mo-Standard (**A**); cell counts of U87 cells treated with 50 μM CoCl_2_ and 7.5 μM Mo-Standard vs. treated with 50 μM CoCl_2_ and 7.5 μM Mo-Glut3 or Mo-PDK-1 (**B**); cell counts of U87 cells treated with 50 μM CoCl_2_ and 2 Gy vs. treated with 50 μM CoCl_2_, 2 Gy, and 7.5 μM Mo-Glut3 or Mo-PDK-1. On the abscissa axis, the origin for the hours corresponds to the start of the treatment (**C**). The data shown are representative of three independent experiments and are expressed as the mean ± standard deviation of the mean (SD). The significance level compared to the control sample (control in (**A**), 50 μM CoCl_2_ and 7.5 μM Mo-Standard in (**B**), and 50 μM CoCl_2_ and 2 Gy in (**C**)), for each time point, was set to *p* < 0.05 (unpaired *t*-test Welch corrected) and displayed with the asterisk (*).

**Figure 3 ijms-25-02079-f003:**
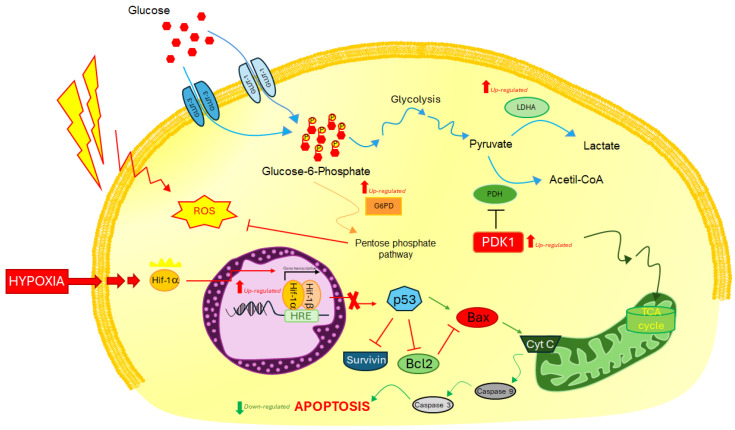
Pathways related to the selected key genes involved in the induction of Warburg effect (*glut-1*, *glut-3*, *ldha*, and *pdk-1*) and the survival/death balance (*survivin*, *bax*, *bcl-2*, and *casp-9*).

**Figure 4 ijms-25-02079-f004:**
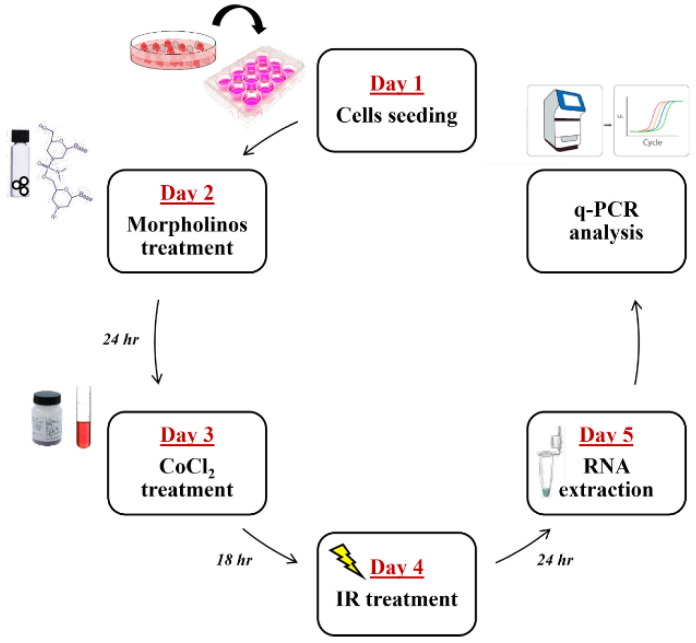
Experimental planning from the cells seeding, on day 1, until the RNA extraction, on day 5, and subsequent qRT-PCR analysis.

**Figure 5 ijms-25-02079-f005:**
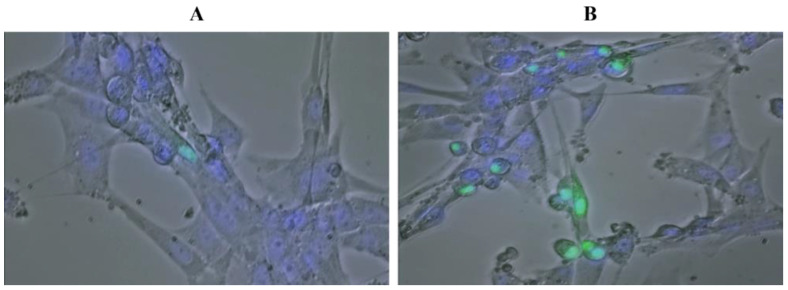
U87 cells transfected with 1 µL/mL of Lipofectamine and 0.8 µg/mL of pEGFP-N1-Hif-1α, untreated (**A**) and treated with CoCl_2_ (**B**) at a concentration of 50 µM. Nuclei staining with 1.5 μg/mL Hoechst (magnification 40×).

**Table 1 ijms-25-02079-t001:** Fold change values of up- and down-regulated genes in U87 cells treated with CoCl_2_, 2 Gy, and CoCl_2_ + 2 Gy.

		Gene Expression Variations by qRT-PCR		
		CoCl_2_	2 Gy	CoCl_2_ + 2 Gy
HRE regulated genes	**hif-1α**	1.05	1.22	1.38
	**glut-3**	4.05	0.70	1.62
	**glut-1**	2.10	0.94	1.92
	**eno-1**	1.78	0.79	1.37
	**ldha**	1.43	1.08	1.41
	**pdk1**	2.06	0.96	2.59
	**p21**	2.16	1.29	2.38
	**lon**	1.30	1.07	1.16
	**lincRNA-p21**	1.5	0.73	0.60
	**mir-210**	1.36	0.99	1.01
	**mir-590**	3.51	1.04	2.03
	**survivin**	0.38	0.71	0.52
	**livin**	0.76	0.49	0.48
Genes indirectly regulated by HIF-1α	**bax**	1.43	1.38	0.95
	**casp-9**	0.7	0.95	0.68
	**bcl2**	1.01	0.16	1.20

**Table 2 ijms-25-02079-t002:** Fold change values of up- and down-regulated genes in U87 cells treated with Mo-PDK-1, Mo-PDK-1 + CoCl_2_, Mo-PDK-1 + CoCl_2_ + 2 Gy, Mo-GLUT3, Mo-GLUT3 + CoCl_2_, Mo-GLUT3 + CoCl_2_ + 2 Gy.

	Gene Expression Variations by qRT-PCR							
	MoPDK1	MoPDK1 + CoCl_2_	MoPDK1 + 2 Gy	MoPDK1 + CoCl_2_ + 2 Gy	MoGLUT-3	MoGLUT-3 + CoCl_2_	MoGLUT-3 + 2 Gy	MoGLUT-3 + CoCl_2_ + 2 Gy
**glut-3**	0.68	0.83	1.32	1.32	-	-	-	-
**glut-1**	2.00	4.92	4.19	8.82	0.74	2.91	1.66	4.53
**pdk1**	-	-	-	-	0.67	1.84	1.02	0.92
**Ldha**	1.21	2.31	1.83	2.34	0.69	1.50	1.51	1.32
**Bax**	0.90	1	1.28	1.38	0.74	0.95	0.90	1.60
**casp-9**	0.72	1.07	1.72	1.05	0.76	1.65	1.5	1.63
**bcl2**	1.64	2.05	1.64	2.30	1.65	1.68	1.93	3.15
**survivin**	1.51	1.1	1.33	1.75	1.32	1.41	1.52	2.34

**Table 3 ijms-25-02079-t003:** Oligonucleotides sequences.

Gene Symbol	Gene Name	Forward Primer 5′>3′	Reverse Primer 5′>3′
** *hif-1α* **	Hypoxia Inducible Factor 1, Subunit α	GGGGGGCTAGCATGAACGACAAGAAAAAGATAAGT	GGGGGGGATCCTTAACTTGATCCAAAGCTCTG
** *hif-1α* **	Hypoxia Inducible Factor 1, Subunit α	GCTTGCTCATCAGTTGCCAC	ATCCAGAAGTTTCCTCACACG
** *glut-1* **	Glucose transporter, Type 1	CATCTTCACTGTGCTCCTGG	CCTCGGGTGTCTTGTCACTT
** *glut-3* **	Glucose transporter, Type 3	CACTTTGCTCTGGGTGGAAG	TCACTGACAAGGGTTTGGCTA
** *eno-1* **	Enolase 1, (Alpha)	TTCGCCCGCACCACTACAG	AGAGCCGTCACTCATTCCCT
** *Ldha* **	Lactate Dehydrogenase A	CAGCCCGATTCCGTTACCTA	TCTTCAGAGAGACACCAGCAA
** *pdk1* **	Pyruvate Dehydrogenase Kinase 1	AACCAAAGCATCAGAGCCATC	TTGAGCCCAGAAGATTGAAGC
** *Livin* **	Baculoviral IAP repeat containing 7	CTGGGCATATTCTGAGATTGG	AGGCACTTGGCACTGTCTTTA
** *survivin* **	Baculoviral IAP repeat containing 5	TCTAAGTTGGAGTGGAGTCTG	CAGTTTGGCTTGCTGGTCTC
** *p21* **	Cyclin dependent kinase inhibitor 1A	CGGAACAAGGAGTCAGACATT	CGTTAGTGCCAGGAAAGACAA
** *Lon* **	lon peptidase 1, mitochondrial	CTGGAGAAGGACGACAAGGA	GGTAGTTGCGGGTGACATTG
** *lincRNA-p21* **	P53 pathway corepressor 1 protein tumor	GGTGGGGCTGAAGTTTATGC	CACACACAGGTGGGTTGATG
** *miR-210* **	microRNA 210	CAGCCCCTGCCCACCGC	TGCCCAGGCACAGATCAGC
** *miR-590* **	microRNA 590	AAATGAGCTTATTCATAAAAGTGC	GCATGTTTCAATCAGAGACTAG

**Table 4 ijms-25-02079-t004:** Morpholinos and related primers sequences.

Morpholino Symbol	Morpholino Sequences 5′>3′	Forward Primer 5′>3′	Reverse Primer 5′>3′
**Mo-GLUT3**	GCCCAGTTTCTAGTCAATACCTGCC	GGTCATCAATGCTCCTGAG	TTCCAACAACGATGCCCAG
**Mo-PDK1**	ACACAAGATGAGAATCTTACCAGCT	ATGAAGCAGTTCCTGGACTT	GCTGATTGAGTAACATTCTAA
**Mo-ST**	CCTCTTACCTCAGTTACAATTTATA	-	-

## Data Availability

Data is contained within the article and Appendix A.

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
