# Peer review of "Glut-3 Gene Knockdown as a Potential Strategy to Overcome Glioblastoma Radioresistance"

_ijms, 2024, doi:10.3390/ijms25042079_

Round 1

Reviewer 1 Report

Comments and Suggestions for Authors

In this manuscript, the authors verified that glut-3 and pdk1knock-down can control the anaerobic use of pyruvate and reduce proliferation rate to overcome radioresistance. However, the research design should be more rigorous. Considering the confusion of some data in this article, I may recommend the rejection of this manuscript. Also, the language is poor and needs to be embellished. Meanwhile, to make the manuscript more complete, there are some suggestions as follows:

1. The authors used CoCl2 to induce chemical hypoxia. CoCl2 stabilize HIF‐1α and affect glycolysis pathway. However, it has been observed that the cell metabolism in low oxygen‐induced hypoxia and CoCl2‐chemically induced hypoxia is different (J Appl Toxicol. 2019;39:556–570). We suggest that authors should add hypoxia chamber to mimic low oxygen‐induced hypoxia condition rather than CoCl2‐chemically induced hypoxia.

2. In Figure 1 showed the cytotoxicity of CoCl2. It was found that 50 mM and 100 mM of CoCl2 exhibited high cytotoxicity after 48 h and 72 h treatment. However, in Figure 2, the authors showed the cell viability of U87-MG at 1 week, 11 days and 15 days. We wonder that U87-MG cells can survive in this condition (with 50 mM of CoCl2) more than 15 days.

3. In this manuscript, they used CoCl2 to mimic hypoxia condition. However, the cells under CoCl2‐chemically induced hypoxia condition cannot be regarded as radioresistant cells. The authors should provide experiments to prove it.

4. In this manuscript, the authors mentioned that under CoCl2‐chemically induced hypoxia, U87-MG favored glycolysis pathway and some glycolysis related gene were high expression (Table 1). It may just a compensation of energy. We suggest the authors should also analysis mitochondrial respiration related gene expression to confirm the metabolic shift from the oxidative phosphorylation to anaerobic glycolysis.  

5. In Figure 2C, the triple combinations (COCl2, Mo-GLUT3 and 2 Gy) showed lower cell survival rate than control group ((COCl2 and 2 Gy). However, in Table 2, the triple combinations group exhibited the highest pro-survival gene (bcl2 and surviving) expression. Please give detailed explanation.

6. The mRNA expression cannot be regarded as protein expression. We suggest the authors should do Western blot experiment to confirm the final HRE regulated protein expression.

7. Some results are not meaningful and does not need to be described in detail, thus it is recommended that the relevant results be placed in the supplementary information. For example, Figure 5 and Figure 7.

Comments on the Quality of English Language

The language is poor and needs to be embellished.

Author Response

We thank the Reviewer for these comments and constructive criticisms.

Reply 1. We agree with the Reviewer that physic hypoxia should have been the most appropriate model for our study. We also previously used hypoxic chambers to conduct the study Bravatà V. et al. J Pers Med. 2021, Apr 16;11(4):308. However, in this particular case, it would have been very complex to manage chambers, due to the large number of biological samples and technical replicates. However, the use of CoCl2 is a consolidated method, which reproduces many metabolic aspects induced by low oxygen concentration. As the article cited by the Reviewer suggests in the conclusion section, the use of CoCl2 must be evaluated in the specific project, by checking the obtained effects on the function and phenotype of the particular cells being studied. We are confident that our model works as we desidered because we obtained  activation of HRE regulated genes and particularly GLUT1-3 and PDK1 key genes, which are  expected to be activated under physical hypoxia.

Reply 2. The figure 1 showed cell cytotoxicity results obtained after 48 and 72 hours post treatment with 50 mM and 100 mM of CoCl2. For combined treatments (figure 2), we decided to treat cells with 50 mM of CoCl2 only for 18 hours, in order to reduce cytotoxic effects, as reported in the manuscript (section Results lane 132 and section Material and Methods lane 488). In particular, within the time window of 18 hours of incubation with CoCl2, cells were irradiated. After irradiation, the medium was changed and cells were left in culture to grow under standard conditions to evaluate viability after 1 week, 11 and 15 days. In order to further clarify this experimental procedure, we have added the following sentence in the Material and method section (lane 489-491): “To evaluate cell proliferation, after 18 hours of combined treatments, medium was changed, cells were left in culture to grow under standard conditions and cell counting was performed at 1 week, 11 and 15 days”.

Reply 3.  Figure 2 shows the proliferation rate observed after 1-15 days after treatments. As shown, the 50 μM CoCl2 sample shows similar cell counts of untreated cells even after 15 days after treatments. Moreover, a recent published article reproduced our studying model and confirmed that simulating hypoxia with CoCl2 effectively increases radioresistance of U87 cells (Khakshour E. et al. https://doi.org/10.1016/j.mrfmmm.2023.111848) .

Reply 4. The aim of this study was to identify key biomarkers of response to hypoxia and irradiation combined treatments. Thus, we focused our attention only on HRE related genes, in order to be sure of contrasting the HIF-1 pathway and reverting the use of glycolysis as a mechanism of anaerobic adaptation. Even if mitochondrial respiration were still activated, it would not add further information on the biomarker to choose  to radio-sensitize cells.

Reply 5. Gene expression results by qRT-PCR are obtained 24 hours post combined treatments. Both the pdk1 and glut-3 genes silencing induced an early expression of pro-survival genes, as cell survival tentative. This aspect is reported in detail in the Result section (lanes 254-268) and Discussion section (lanes 417-429) of the manuscript. As regards to the effects of the triple combination on cell survival, we observed a radiosensitizing effect at later times (7-15 days), a result expected following the combined treatments and explained in the aforementioned Discussion section.

Reply 6. Western blot assays would surely be a useful in-depth analysis to highlight the HRE regulated genes modifications pre-and post the morpholino/IR single or combined treatments. In the initial experimental design of this study, we retained more appropriate to conduct gene expression assays, as  a gene silencing study approach was adopted. However, as the final aim of this study is the U87 radiosensitization, we retained more information about the late effects induced on the proliferative rates at 1-2 weeks post treatments. Indeed, gene (or protein) modifications are early changes, which offer molecular explanations of the reduced proliferation ability after GLUT-3/PDK1 gene silencing. We have already developed an in-vivo GBM xenograft model (authorization n° 970/2018-PR from Italian Health Ministry), where we’re studying the GBM hypoxic areas with imaging approach. Thus, we will test the Mo-Glut3 on this preclinical model and verify the GLT-3 protein knock down by immuno-istochimical approach.  

Reply 7. Figure 5 and figure 7 were placed in the supplementary information section, as supplementary file 1 and 2, respectively.

Reply 8. We rewrote some sentences and performed English editing to improve readability by using the software Grammarly.

Reviewer 2 Report

Comments and Suggestions for Authors

1. Can you provide more details about the specific methods used for gene knock-down of gllut-3 and pdk-1, such as the concentration and delivery system.

2. How do the findings of this study contribute to our understanding of the molecular response to hypoxia and radioresistance in glioblastoma?

3. What are the limitations of this study, and what future directions do you suggest for further research in this area?

Comments on the Quality of English Language

Quality of English is good. The paper is well written and easy to understand.

Author Response

We thank the Reviewer for these comments and constructive criticisms.

Reply 1. Vivo-Morpholino antisense oligonucleotides were specifically designed to perform in vitro silencing of GLUT3 and PDK1 mRNA. These are oligonucleotides with a unique covalently linked delivery moiety, composed of an octa-guanidine dendrimer. They are completely stable in cells, uncut by nucleases, and very simple to use. Once added to the cell's medium, Vivo-Morpholinos freely diffuse between the cytosol and nuclear compartments and bind complementary sequences of RNA. Therefore, after evaluating their effect in a wider concentration range (2,5-15 μM), we have chosen the 7,5 μM concentration, which showed optimal target silencing. In addition, a Vivo-Morpholino Standard control (GENE TOOLs, LLC), that targets a human beta-globin intron mutation, was used as a control of intrinsic aspecificity and cytotoxicity.

The following sentences were added in the Material and Methods section of the manuscript:

  • “Vivo-Morpholinos freely diffuse between the cytosol and nuclear compartments and bind complementary sequences of RNA” (lanes 545-546)
  • “A vivo Morpholino Standard (MO-ST) oligonucleotide was also purchased and used as control of intrinsic aspecificity and cytotoxicity” (lane 547-548)

Reply 2. Many authors tried to sensitize cancer cells, by inhibiting the Warburg effect. Fewer authors applied this strategy to GBM [Guda MR.et al. DOI: 10.3390/cancers11091308; Velpula, KK. et al. DOI: 10.1158/0008-5472.CAN-13-1868]. In this regard, Vartanian A. and colleagues, targeted hexokinase 2 (HK2) to sensitize U87 and primary GBM cells in a xenograft GBM model subjected to combined radio-chemotherapy with TMZ treatment [Vartanian A. et al. https://doi.org/10.18632/oncotarget.11680]. However, HK2 is also expressed by normal muscle cells, thus systemic toxicity could be expected by developed systemic HK2 inhibitors. Our finding suggests GLUT3 as a knock-down target, alternative to the HK2. The main advantage of GLUT-3 silencing would derive from its higher neuronal expression specificity, whereas in other tissues it is less prevalently expressed respect to the other transporters which could supply the glucose internalization. Thus, the GLUT-3 higher expression in the hypoxic GBM area could represent a way to address the radiosensitization activity specifically on the GBM hypoxic tumor areas (see Discussion section, lanes 433-455).

Reply 3. The study was performed in vitro on the U87 cell line, a widely used human glioblastoma (GBM) cell line in scientific research as a model to understand GBM biology and to develop novel therapeutic strategies for this aggressive form of brain cancer. Our molecular strategy to overcome radioresistance provided promising results that can also be applied in vivo for validation on animal models. Indeed, the transfection system we used in vitro, the Vivo-morpholino, is employed in molecular biology and biomedical research for gene knockdown experiments in in-vivo studies in living organisms. In particular, we have in progress a study approved by the Italian Ministry of Health (authorization n° 970/2018-PR) regarding the impact of hypoxia in GBM in immunodeficient mice by using hadrontherapy treatments and micro-PET molecular imaging. In this context, as future objectives we aim to validate our gene silencing model in in vivo GBM model as a radiosensitization system for radiotherapy treatment.

Reviewer 3 Report

Comments and Suggestions for Authors

This work was developed to identifying key genes upregulated by HIF in U87 cells subjected to radiations under hypoxia condition, in order to test their silencing as a radiosensitization method to overcome radioresistance. In my opinion. The study can be published in this journal after major corrections:

1-The format of the tables should be written according to the guidance

2-Please discuss about data of Figure 4.

3-On what basis are the results of Figure 5 are presented?

4-Please discuss the uniformity of Figure 6.

5- More related literatures should be added, for examples A) .doi: https://doi.org/10.1016/j.jpha.2023.04.015, B) https://doi.org/10.7150/jca.83615, C) https://doi.org/10.5152/AnatolJCardiol.2021.60378 and D) doi: 10.3390/cancers15010026

6- Here novelty of the research work is not clear, elaborate on the novelty of the research work.

7-English writing is very poor.

Comments on the Quality of English Language

English writing is very poor.

Author Response

We thank the Reviewer for these comments and constructive criticisms.

Reply 1. The format of the tables was modified according to the guidelines.

Reply 2. The Figure 4 describes the experimental procedure we used to perform combined treatments from the day of cell seeding to the gene expression investigations by using qrt-PCR. In particular, as we reported in the Material and Methods section of the manuscript (lanes 485-489), the experimental workflow was the following: day 1: U87 cells seeding; day 2: cell treatment with 7.5 μM Mo-ST or 7.5 μM Mo-Glut3 or 7.5 μM MoPDK1; day 3: medium change and treatment with 50 μM CoCl2 for 18 hours; day 4: cell irradiation with 2 Gy X-rays; day 5: total RNA extraction and qRT-PCR.

Reply 3. The Figure 5 shows results obtained on U87 cell lines in order to test the cytotoxicity of the transfection system (Lipofectamine) alone and in combination with CoCl2 to induce chemical hypoxia. In particular, percentages of viable and dead cells are shown in the control and in samples treated after the transfection with 1 μl/ml of Lipofectamine and 0.8 μg/ml of plasmid DNA and after transfection co-treatment with 50 or 100 μM CoCl2. We explained the results of Figure 5 in the Material and Methods section (lanes 518-526). As suggested by the Reviewer 1 this Figure and Figure 7 were placed in the supplementary information section, as supplementary file 1 and 2, respectively.

Reply 4. These images are representative of the transfection experiments performed. We rephrased in the Material and Methods section of the manuscript (lanes 529-532), this sentence as follows: “The chosen combination of 1 μl/ml of transfecting agent and 0.8 μg/ml of plasmid DNA showed a higher number of GFP-positive cells (2,5x) than untreated cells, overall showing higher nuclear GFP fluorescence intensity (AU) (8x)”.

In addition, as requested by the Reviewer 1 Figure 6 has been renumbered as Figure 5.

Reply 5. We added the reference https://doi.org/10.1016/j.jpha.2023.04.015 as it is focused on the topics of the manuscript (lane 435, new reference n° 55).

Reply 6. The novelty of this study is the covering of a literature gap, as few authors tried to sensitize GBM  by inhibiting the Warburg effect [Guda MR.et al. DOI: 10.3390/cancers11091308; Velpula, KK. et al. DOI: 10.1158/0008-5472.CAN-13-1868]. In this regard, Vartanian A. and colleagues,  targeted hexokinase 2 (HK2) to sensitize U87 and primary GBM cells in a xenograft GBM model subjected to combined radio-chemotherapy with TMZ treatment [Vartanian A. et al. https://doi.org/10.18632/oncotarget.11680]. However, HK2 is also expressed by normal muscle cells, thus systemic toxicity could be expected by developed systemic HK2 inhibitors. Our finding suggests GLUT3 as a knock-down target, alternative to the HK2. The main advantage of GLUT-3 silencing would derive from its higher neuronal expression specificity, whereas in other tissues it is less prevalently expressed respect to the other transporters which could supply the glucose internalization. Thus, the GLUT-3 higher expression in the hypoxic GBM area could represent a way to address the radiosensitization activity specifically on the GBM hypoxic tumor areas.

7.English writing is very poor.

Reply 7. We rewrote some sentences and performed English editing to improve readability by using the software Grammarly.

Round 2

Reviewer 1 Report

Comments and Suggestions for Authors

The physical hypoxia condition is difficult to mimic in in vitro experiment. We believe that the authors have done their best to provide relevant information.

However, they just described the result of the triple combination on cell survival ,but they still do not explain it. In general, the triple combined treatment lead to low cell survival rate should be accompanied by high pro-aopototic signals.

Author Response

Reply. We thank the Reviewer for his/her valuable comment and constructive criticism.

As regards our in vitro model of chemical hypoxia, we assessed the correct hypoxia establishment on two levels, directly, by highlighting fluorescent nuclei due to the expression of the recombinant HIF protein from the HIF1a-GFP transgene transfection, and indirectly, by evaluating gene expression activation of its downstream, HRE-regulated target genes. Both these approaches demonstrated the hypoxia instauration in our model of U87 GMB cell line.

We also explained in detail our results about cell survival of triple combination and qRT-PCR results with particular reference to the regulation of the survival/death balance in the Results section (lanes 254-268) and Discussion section (lanes 361-366 and 417-429). As we reported, both the PDK1 and GLUT-3 genes silencing induced an early survival tentative, evaluated a few hours post hypoxia treatment, even if the proliferative trend assessed at 7, 11 and 15 days after triple treatments, showed in both cases significant proliferative rate reduction, as a sign of the radiosensitization ability of these two an-tisense mRNAs.

Overall, our early gene expression results by qrt-PCR are in line and confirmatory of our previous study carried out in the same U87 GBM cell line where we induced physical hypoxia and performed gene expression profiling experiments by microarray analysis 24 hours after hypoxia and treatment with 2 Gy of ionizing radiation (IR) (Bravatà V et al, 2021 https://pubmed.ncbi.nlm.nih.gov/33923454/). Indeed, in that study (reference n°15 in the manuscript) we observed 24 hours post hypoxia and combined treatment with IR, an upregulation of specific pathways related to cell fate (such as p53 signaling, cell cycle), tumor progression, cell–cell communication, angiogenesis and invasiveness), suggesting that the highly aggressive GMB cells are able to respond positively to stress signals. The use of our gene silencing system is therefore effective in overcoming the resistance of these cells at later times.

In light of these considerations, we have added the following sentence in the Discussion section of the manuscript (lines 425-428): “These results regarding the survival/death balance were also in line with gene expression profiling experiments previously performed by our group on the U87 cells subjected to physical hypoxia and 2 Gy of IR [15]”.

Reviewer 3 Report

Comments and Suggestions for Authors

Corrections are acceptable.

Author Response

Reply. We thank the Reviewer for his/her interest and attention in reviewing our manuscript.